# SEFE: Superficial and Essential Forgetting Eliminator for Multimodal Continual Instruction Tuning

Jinpeng Chen [* 1]   Runmin Cong [* 2]   Yuzhi Zhao [1]   Hongzheng Yang [3]
Guangneng Hu [4]   Horace Ho Shing Ip [1]   Sam Kwong [5]
https://github.com/jinpeng0528/SEFE/

## Abstract

Multimodal Continual Instruction Tuning (MCIT) aims to enable Multimodal Large Language Models (MLLMs) to incrementally learn new tasks without catastrophic forgetting. In this paper, we explore forgetting in this context, categorizing it into superficial forgetting and essential forgetting. Superficial forgetting refers to cases where the model's knowledge may not be genuinely lost, but its responses to previous tasks deviate from expected formats due to the influence of subsequent tasks' answer styles, making the results unusable. By contrast, essential forgetting refers to situations where the model provides correctly formatted but factually inaccurate answers, indicating a true loss of knowledge. Assessing essential forgetting necessitates addressing superficial forgetting first, as severe superficial forgetting can obscure the model's knowledge state. Hence, we first introduce the Answer Style Diversification (ASD) paradigm, which defines a standardized process for transforming data styles across different tasks, unifying their training sets into similarly diversified styles to prevent superficial forgetting caused by style shifts. Building on this, we propose RegLoRA to mitigate essential forgetting. RegLoRA stabilizes key parameters where prior knowledge is primarily stored by applying regularization, enabling the model to retain existing competencies. Experimental results demonstrate that our overall method, SEFE, achieves state-of-the-art performance.

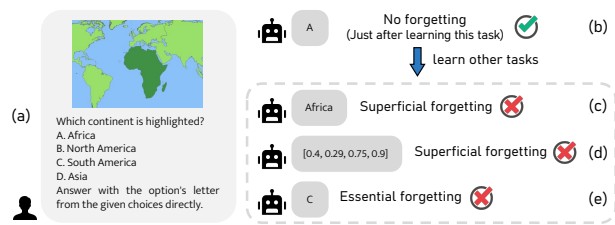

*Figure 1.* Examples illustrating *superficial forgetting* and *essential forgetting*. (a) Instruction; (b) Response case without forgetting; (c) and (d) Response cases with *superficial forgetting*; (e) Response case with *essential forgetting*.

## 1. Introduction

The rapid advancement of Multimodal Large Language Models (MLLMs) has led to a series of significant works (Liu et al., 2023; Dai et al., 2023; Achiam et al., 2023; Bai et al., 2023b; Liu et al., 2024a; Chen et al., 2024d). Training these models typically involves multiple phases, with pre-training and instruction tuning being two crucial ones. During pre-training, large-scale unsupervised learning endows the model with foundational cognitive skills. In the subsequent instruction tuning phase, the model is refined using instruction-response pairs to enhance its ability to comprehend and respond to user commands. This comprehensive training enables MLLMs to exhibit zero-shot capabilities, allowing them to handle previously unseen instructions.

However, these zero-shot abilities are often insufficient for MLLMs to specialize in specific domains. To achieve such specialization, targeted multimodal instruction tuning with relevant datasets is commonly employed. Nevertheless, the breadth of tasks that MLLMs are expected to handle is vast, and the requirements are continuously evolving. As demands grow, simply fine-tuning the model with data from new tasks can result in the loss of previously acquired capabilities—a phenomenon known as catastrophic forgetting (McCloskey & Cohen, 1989). Alternatively, retraining the model with both new and old task data each time requirements change is highly resource-intensive. To address this, Multimodal Continual Instruction Tuning (MCIT) has

---
[*]Equal contribution [1]City University of Hong Kong [2]Shandong University [3]The Chinese University of Hong Kong [4]Xidian University [5]Lingnan University. Correspondence to: Yuzhi Zhao <yzzhao2-c@my.cityu.edu.hk>, Sam Kwong <samkwong@lingnan.edu.hk>.

*Proceedings of the 42nd International Conference on Machine Learning*, Vancouver, Canada. PMLR 267, 2025. Copyright 2025 by the author(s).

emerged (He et al., 2023; Zhu et al., 2024a; Chen et al., 2024a; Zheng et al., 2024; Zeng et al., 2024). This field seeks to enable MLLMs to progressively learn new tasks without losing proficiency in previously mastered ones during incremental instruction tuning.

Existing methods in MCIT (He et al., 2023; Zhu et al., 2024a; Zheng et al., 2024), like conventional continual learning research, conceptualize catastrophic forgetting as a generalized problem of knowledge loss and have made some progress addressing it. However, our analysis indicates that the nature of forgetting in MCIT extends beyond mere knowledge loss. We propose to categorize it into two types: *superficial forgetting* and *essential forgetting*. *Superficial forgetting* refers to cases where the model's response style deviates from expected norms after learning new tasks. For instance, as shown in Fig. 1(a), a multiple-choice question task (Lu et al., 2022) may prompt the model to "answer with the option's letter". If subsequent tasks demand different response formats—such as certain tasks that require word-based answers (Singh et al., 2019)—the model may begin answering multiple-choice questions with words instead, as shown in Fig. 1(c). Similarly, if later tasks involve bounding box-based grounding (Kazemzadeh et al., 2014; Mao et al., 2016), the model might respond to multiple-choice questions with bounding boxes, as illustrated in Fig. 1(d). On the other hand, *essential forgetting* refers to the actual loss of knowledge, where the model answers in the correct format but with incorrect content, as depicted in Fig. 1(e). Accurately evaluating *essential forgetting* necessitates addressing *superficial forgetting*, since the latter can obscure the model's true knowledge state, making it difficult to determine if knowledge has truly been lost. Nevertheless, current methods fail to recognize these two types of forgetting or their interrelationship, leading to suboptimal performance when applied to realistic benchmark datasets featuring diverse answer formats (Chen et al., 2024a).

We posit that the main cause of *superficial forgetting* is the bias introduced by using a single question format per task. When a model is trained over consecutive batches of questions in one specific style, it becomes biased towards that style, making it difficult or even impossible to respond in other styles. To address this, we propose the Answer Style Diversification (ASD) paradigm. Specifically, after examining 15 prevalent MLLM benchmarks (Young et al., 2014; Goyal et al., 2017; Gurari et al., 2018; Mishra et al., 2019; Singh et al., 2019; Agrawal et al., 2019; Lu et al., 2022; Schwenk et al., 2022; Li et al., 2023c;a; 2024a;b; Liu et al., 2024c; Yu et al., 2024b; Fu et al., 2024), we identify five question types: yes/no questions, multiple-choice questions, short answer questions, brief explanation/description questions, and detailed explanation/description questions. These types also represent the major application scenarios for MLLMs. Based on this observation, we propose refor-

matting each task's dataset into these five styles (the original and four alternatives). This allows the model to generate responses in multiple styles on the same topic during training, thereby mitigating bias from single-format training. Our experiments demonstrate that converting only 10% of the dataset into four alternative styles (2.5% each) substantially reduces *superficial forgetting* and enhances performance.

After addressing *superficial forgetting*, the remaining challenge centers on knowledge loss, namely *essential forgetting*. Our analysis of model parameter changes reveals that certain parameters encode significantly more critical historical knowledge than others. Based on this insight, we introduce RegLoRA, an extension of LoRA (Hu et al., 2022) designed to minimize knowledge loss. RegLoRA identifies crucial elements within the weight update matrices of prior LoRAs and applies regularization to these elements when training new LoRAs. This process stabilizes the subset of parameters closely tied to prior knowledge, thereby effectively preserving acquired proficiency. Furthermore, since the regularized elements constitute only a small fraction of each weight update matrix, and the weight update matrix is the product of parameter matrices $A$ and $B$ rather than the parameters themselves, the constraints on parameter updates are minimal. This focused and restrained approach allows the model to preserve previous knowledge while maintaining the flexibility to learn new information.

Building upon ASD and RegLoRA, we introduce a novel MCIT method called the Superficial and Essential Forgetting Eliminator (SEFE). Our contributions are as follows:

- To our knowledge, we are the first to formally define *superficial forgetting* and *essential forgetting* in MCIT. Furthermore, our proposed method, SEFE, addresses these challenges and achieves state-of-the-art performance.

- To mitigate *superficial forgetting*, we introduce the ASD paradigm that unifies the answer domain across tasks by rephrasing questions, thereby reducing the model's bias toward specific response styles. Additionally, we create CoIN-ASD, an ASD-adjusted version of the CoIN benchmark, which can serve as a new benchmark for evaluating *essential forgetting* in future MCIT studies.

- To address *essential forgetting*, we present RegLoRA. By identifying critical elements in the weight update matrices and applying regularization constraints, RegLoRA ensures that LoRA fine-tuning does not disrupt the model's existing knowledge.

## 2. Related Work

### 2.1. Continual Learning

Most traditional continual learning methods target straightforward tasks like image or text classification (Ke & Liu, 2022; Wang et al., 2024a). These methods are commonly categorized into three types: regularization-based, replay-based, and parameter isolation-based approaches. Regularization-based methods (Dhar et al., 2019; Douillard et al., 2020; Yu et al., 2024c; Luo et al., 2024; Chen et al., 2024c) constrain parameter updates to retain prior knowledge. Replay-based methods (Rebuffi et al., 2017; de Masson D'Autume et al., 2019; Ostapenko et al., 2019; Chen et al., 2023; Luo et al., 2025; Chen et al., 2025) store or generate some past data and replay it during subsequent training to recall earlier information. Parameter isolation-based methods (Mallya & Lazebnik, 2018; Geng et al., 2021; Wang et al., 2022) introduce new parameters to capture additional information while keeping existing parameters unchanged, thereby accommodating both past and newly acquired knowledge.

The proposed ASD paradigm deviates from these established categories. Instead, it is a data reconstruction-based technique, which we believe represents a promising new paradigm for continual learning in the context of modern large-scale models. On the other hand, the proposed RegLoRA can be considered as a regularization-based technique. However, unlike traditional methods that often require auxiliary models or significant extra parameters deployed on GPUs, RegLoRA is better suited for resource-intensive MLLMs.

### 2.2. Continual Learning for LLMs and MLLMs

Recently, continual learning research for LLMs and MLLMs (Wu et al., 2024b; Shi et al., 2024) has grown significantly. Some methods focus on continual pre-training (Gupta et al., 2023; Wu et al., 2024a; Chen et al., 2024b), while others emphasize continual instruction tuning (Yang et al., 2024; Dou et al., 2024; Yu et al., 2024a; Zeng et al., 2024; Cao et al., 2024) as in our approach. For example, O-LoRA (Wang et al., 2024b) prevents overwriting prior knowledge by enforcing orthogonality among LoRA weights across tasks. LoTA (Panda et al., 2024) employs a two-stage training process to identify parameters necessitating updates and modifies only these parameters, thus minimizing model changes. EProj (He et al., 2023) introduces task-specific projection layers and leverages task similarities to enable module sharing, reducing forgetting. Model Tailor (Zhu et al., 2024a) identifies sensitive and salient patches within the new model, decorates them, and integrates them with the previous model, thereby minimizing alterations to the existing model. SAPT (Zhao et al., 2024) employs shared attention to align the learning and selection of parameter-

efficient tuning, mitigating catastrophic forgetting and promoting knowledge transfer.

Most of these approaches, similar to traditional continual learning methods, address catastrophic forgetting as a single challenge. However, this strategy may not be suitable for modern large-scale models, which are inherently larger and entail more intricate training processes. Hence, these models may exhibit forgetting patterns that are distinct from those observed in smaller-scale models studied in traditional continual learning works. To address this, we propose to decompose forgetting in MCIT into two types: *superficial forgetting* and *essential forgetting*. By developing targeted strategies for each type, our method achieves superior performance.

## 3. Answer Style Diversification Paradigm

In the incremental training process of MCIT, each task typically follows a single question format, which may bias the model toward a uniform response style. Consequently, the model tends to answer all questions in this single manner, creating challenges when handling previously learned tasks with different question formats. This limitation results in what we term *superficial forgetting*.

To address this issue, we introduce the ASD paradigm. ASD converts each task's single-format instruction-tuning data into a multi-format dataset with five question types: yes/no, multiple-choice, short answer, brief explanation/description, and detailed explanation/description. As exemplified in Fig. 2, the original dataset for a task may contain only short answer questions. In the ASD paradigm, we transform $X\%$ of the samples equally across the four alternative formats ($\frac{X}{4}\%$ each) and retain the remaining $(100 - X)\%$ in their original formats. This configuration ensures that the final dataset includes all five question types without altering the total sample size, thus avoiding extra influencing factors. For these conversions, we establish a standardized transformation process, mainly leveraging existing MLLMs, with some direct conversions handled through fixed rules. Additionally, following (Liu et al., 2024a), we add a response format prompt (RFP) for each question type.

### 3.1. Question Types and Transformation Process

**Yes/No questions** are those that can be answered with *"Yes"* or *"No"*. To convert other question types into this format, we apply the RFP *"Is the answer correct? Answer 'Yes' or 'No'."* The updated instruction is structured as "original question + a potentially correct answer + RFP". For half of the samples, we include the correct answer, *i.e.*, the original ground-truth (GT) label, as the potentially correct answer and set the new GT label to *"Yes"*. For the remaining half, we generate a plausible but incorrect answer using MLLMs

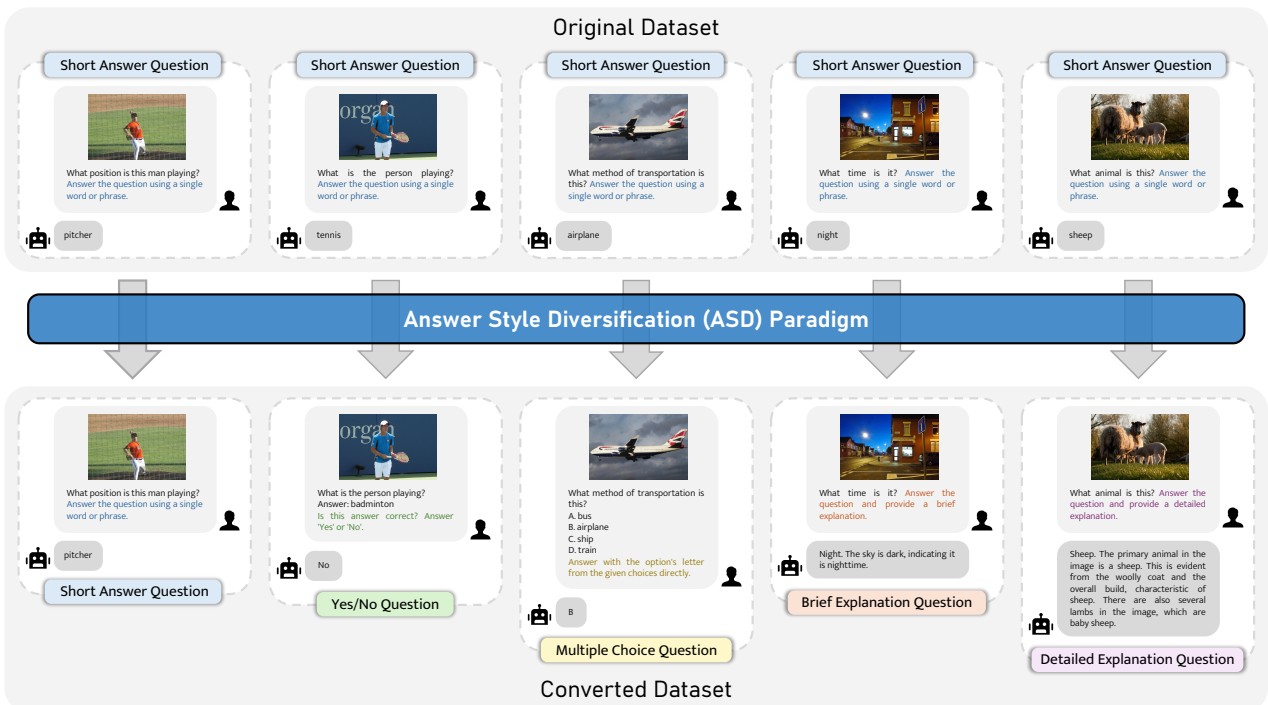

*Figure 2.* An example of the ASD paradigm applied to a dataset consisting solely of short answer questions. Through the ASD process, $(100 - X)\%$ of samples are retained in their original form, while the remaining $X\%$ are equally transformed into four alternative formats: yes/no, multiple-choice, brief explanation, and detailed explanation. Additional examples are provided in Appendix B.

and set the GT label to *"No"*.

**Multiple-choice questions** are characterized by several answer options and utilize the RFP *"Answer with the option's letter from the given choices directly"*. To convert other question types into a multiple-choice format, we structure the instruction as "original question + option list + RFP". If the original question is a yes/no question, the option list includes *"A. Yes"* and *"B. No"*. For other original question types, MLLMs are used to generate three plausible distractor options, which are then shuffled with the correct answer to create a four-option list. The GT label is then set to the letter corresponding to the correct option.

**Short answer questions** correspond to concise and direct GT responses, typically consisting of a single word or phrase. The RFP for these questions is *"Answer the question using a single word or phrase"*, with the modified instruction structured as "original question + RFP". We consider three cases during the conversion process: (1) If the original question is a yes/no question, half of the samples to be converted retain their GT labels of *"Yes"* or *"No"*, as these can be considered as short answers. To avoid biasing the model toward yes/no responses in short answer format, the remaining half are transformed using MLLMs to generate similarly focused questions that require answers other than *"Yes"* or *"No"*, replacing the original question

and GT label. (2) For multiple-choice questions, the content of the correct option becomes the new GT label. (3) For questions with brief or detailed explanations/descriptions, we use MLLMs to condense the GT label into a short answer of no more than ten words.

**Brief explanation/description questions** require answers of about 20 words, presented as concise explanations or descriptions. The instruction structure is also "original question + RFP". For original questions with direct answers, *e.g.*, yes/no, multiple-choice, or short answer questions, the RFP is *"Answer the question and provide a brief explanation"*. The new GT label consists of the direct answer followed by a roughly 20-word explanation generated by MLLMs. For original questions without a direct answer, such as descriptive or detailed explanation questions, the original answer is reformulated by MLLMs into an approximately 20-word GT label. The RFP for this scenario is *"Answer the question using a brief explanation/description"*.

**Detailed explanation/description questions** are similar to the brief counterparts but require more comprehensive responses of approximately 50 words. The conversion process mirrors that of the brief explanation/description questions. The corresponding RFP is *"Answer the question and provide a detailed explanation"* or *"Answer the question using a detailed explanation/description"*.

Our review of prevalent MLLM benchmarks (Young et al., 2014; Goyal et al., 2017; Gurari et al., 2018; Mishra et al., 2019; Singh et al., 2019; Agrawal et al., 2019; Lu et al., 2022; Schwenk et al., 2022; Li et al., 2023c;a; 2024a;b; Liu et al., 2024c; Yu et al., 2024b; Fu et al., 2024) and common MLLM application scenarios suggests that the five question formats described above can adequately address the majority of MCIT contexts. While certain cases may require minor adjustments, we believe that our standardized process provides a strong foundational framework adaptable to various MCIT applications. By applying this framework to data transformation, *superficial forgetting* can be effectively mitigated, thereby significantly boosting performance. Our experiments demonstrate that transforming as little as 10% of the data (*i.e.*, $X = 10$) yields substantial improvements. Further details are provided in Appendix B.

### 3.2. CoIN-ASD Benchmark

CoIN (Chen et al., 2024a) is the first public benchmark for MCIT, incorporating eight tasks with diverse question formats. Assessing *essential forgetting* on CoIN is challenging due to the frequent occurrence of *superficial forgetting*. Although CoIN includes evaluations using a large language model (LLM) (Bai et al., 2023a) to measure knowledge capability, which can be useful in some contexts, this approach has limitations in certain extreme cases. For instance, as illustrated in Fig. 1(d), the model's response to a multiple-choice question is a bounding box, making it impossible for the LLM to assess the knowledge state. To address this limitation, we apply ASD to the CoIN dataset, creating an updated version we term CoIN-ASD. Training models on CoIN-ASD reduces *superficial forgetting*, allowing future MCIT methods to focus on knowledge retention and *essential forgetting*. In developing CoIN-ASD, we utilized InternVL2-26B (OpenGVLab Team, 2024). CoIN-ASD provides various versions with $X$ values of 10, 20, 40, 60, and 80, facilitating a more comprehensive analysis.

### 4. RegLoRA

After addressing *superficial forgetting*, we shift our focus to *essential forgetting*. To mitigate this, we introduce RegLoRA, a variant of LoRA (Hu et al., 2022) optimized to minimize *essential forgetting*, as illustrated in Fig. 3. During training with RegLoRA, each task is learned by fine-tuning the model with LoRA. Upon completing a task, we identify the key elements in LoRA's weight update matrix that are critical to the knowledge acquired for that task. This LoRA is then merged into the main model, and a new LoRA is initialized for the next task. During subsequent training of new LoRAs, regularization is applied to all previously identified key elements, preserving essential prior knowledge.

### 4.1. Background: LoRA

LoRA (Hu et al., 2022) is a widely used parameter-efficient fine-tuning (PEFT) technique designed to match or even surpass the performance of full-parameter fine-tuning while requiring significantly fewer resources. It achieves this efficiency by modeling the weight updates of the neural network's linear layers using low-rank matrices, thereby reducing the number of trainable parameters.

Specifically, consider a weight matrix $W \in \mathbb{R}^{k \times d}$ in a linear layer, where $d$ and $k$ denote the input and output dimensions. LoRA introduces two low-rank matrices, $A \in \mathbb{R}^{r \times d}$ and $B \in \mathbb{R}^{k \times r}$, to model the weight updates $\Delta W \in \mathbb{R}^{k \times d}$. The modified weight matrix during fine-tuning is:

$$W' = W + \Delta W = W + B \times A, \tag{1}$$

where $r \ll \min(d, k)$ ensures $A$ and $B$ have significantly fewer parameters than $W$. By updating only $A$ and $B$ while keeping $W$ fixed, LoRA achieves high parameter efficiency.

### 4.2. Key Elements Identification

Since we initialize a LoRA for each task, this LoRA captures the incremental knowledge between the new task and the model's existing capabilities. As shown in Eq. 1, this knowledge is encoded in the model parameters as a weight update matrix, $\Delta W = B \times A$. Our analysis of these weight update matrices during incremental training reveals that the updates are not uniformly distributed across all elements. Specifically, some elements have significantly larger absolute values, while others are much smaller. For instance, the average absolute value of the top 1% of elements is 580 times greater than that of the bottom 1%. This finding suggests that preserving task memory relies heavily on stabilizing the parameters associated with the large elements in the weight update matrix, as these elements more critically capture the incremental knowledge required for the task.

To achieve this, we first identify these key elements in the LoRA's weight update matrix based on their large absolute values. Specifically, after completing the $i$-th task, we compute the weight update matrix $\Delta W_i$ for LoRA in each linear layer by $\Delta W_i = B_i \times A_i$. We then select the top $M\%$ of elements based on their absolute values. Once the LoRA is merged into the main model, parameters corresponding to these selected positions undergo notable changes to reflect newly acquired knowledge. Thus, to retain knowledge from the current task during subsequent fine-tuning, it is crucial that the parameters at these positions remain unchanged. To enforce this, we construct a regularization mask $R_i \in \mathbb{R}^{k \times d}$, assigning a value of $1$ to these positions and $0$ to others. This mask records the positions that should remain unaltered during subsequent training. By creating these regularization masks, we establish a foundation for preserving essential knowledge in future training.

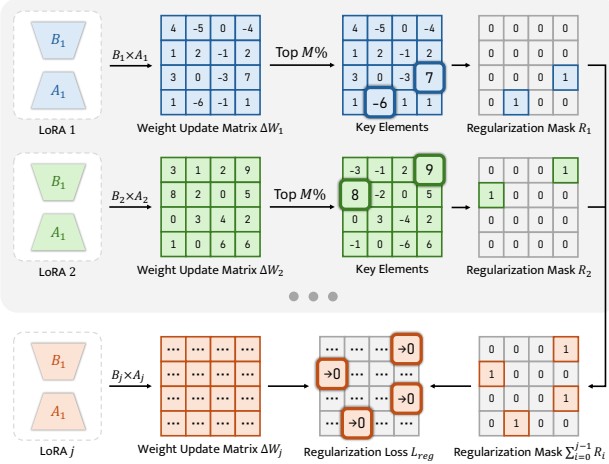

*Figure 3.* Overview of RegLoRA. In each past LoRA, large values in the weight update matrix are identified as key elements. When training a new LoRA, these key positions are incorporated into a regularization mask to enforce targeted constraints.

### 4.3. Regularized Training

In the training of subsequent LoRAs for future tasks, we introduce a regularization loss that utilizes all previous regularization masks to ensure effective knowledge retention. Specifically, during the training of the $j$-th task, the regularization loss is defined as:

$$\mathcal{L}_{reg} = \lambda \sum_{i=1}^{j-1} |\Delta W_j| \otimes R_i, \qquad (2)$$

where $\lambda$ is a balancing hyperparameter, $\otimes$ denotes the Hadamard product, and $\Delta W_j = B_j \times A_j$ is the weight update matrix of the new LoRA. During training, $\mathcal{L}_{reg}$ is added to the original loss of the base MLLM to form the complete learning objective. It aggregates the positions indicated by prior regularization masks. By optimizing this loss, the elements at these positions are encouraged to approach zero, thereby preserving parameters that encode essential prior knowledge. Typically, regularization masks from different past tasks rarely assign 1 to the same position, as elements marked by earlier masks tend to zero during the learning of subsequent tasks. Thus, elements at these positions in later LoRAs are less likely to become key elements. In rare cases of overlap, the regularization weight of these elements in $\mathcal{L}_{reg}$ accumulates, as defined in Eq. 2, which increases resistance to further updates.

This strategy focuses on the top $M\%$ of key elements that embody essential prior knowledge for regularization, effectively mitigating *essential forgetting*. At the same time, most elements remain unrestricted, allowing the model to adapt to new information with minimal limitations. Notably, even

for elements under regularization, it is not the parameters but the dot products of associated vectors in $B_j$ and $A_j$ that are regulated. This permits flexibility across all parameters in $B_j$ and $A_j$, as long as certain dot products stay near zero. These characteristics enable RegLoRA to achieve the dual objectives of retaining prior knowledge and assimilating new information.

## 5. Experiments

### 5.1. Benchmark and Evaluation Metrics

We conduct evaluations using the public MCIT benchmark, CoIN (Chen et al., 2024a), which comprises eight widely used vision-language tasks: ScienceQA (SQA) (Lu et al., 2022), TextVQA (VQA$^{\text{Text}}$) (Singh et al., 2019), ImageNet (ImgNet) (Deng et al., 2009), GQA (Hudson & Manning, 2019), VizWiz (Gurari et al., 2018), Grounding (Grd) (Kazemzadeh et al., 2014; Mao et al., 2016), VQAv2 (VQA$^{\text{v2}}$) (Goyal et al., 2017), and OCR-VQA (VQA$^{\text{OCR}}$) (Mishra et al., 2019). The evaluation framework of CoIN consists of two components: Truth Alignment (TA), which assesses exact matches between answers and GT labels, and Knowledge Capability (KC), which leverages Qwen1.5-32B (Bai et al., 2023a) to evaluate the correctness of knowledge presented in answers. Additionally, we use CoIN-ASD, an ASD-adjusted version of CoIN, to evaluate *essential forgetting* in both our method and existing methods. Although KC partially reflects *essential forgetting*, it does not account for cases where *superficial forgetting* fully obscures knowledge retention in answers, *e.g.*, the example in Fig. 1(d). This limitation makes CoIN-ASD a necessary addition to our evaluation. Given the constraints of KC, we prioritize TA in our evaluation, with KC results provided in Appendix E.

For both TA and KC, we report the accuracy of each learned task after the model has been trained on all tasks. Additionally, we present four aggregate metrics: (1) **Mean Fine-tune Accuracy (MFT)**, the average accuracy for each task immediately after it is learned, representing an upper-bound performance without forgetting. (2) **Mean Final Accuracy (MFN)**, the average accuracy across all tasks after the complete incremental training sequence, reflecting the model's ultimate performance. (3) **Mean Average Accuracy (MAA)**, the mean of the average accuracies on all learned tasks after each incremental training step, providing a comprehensive measure of performance throughout the training process. (4) **Backward Transfer (BWT)**, the difference in accuracy for each task between after all tasks have been learned and immediately after learning that task, assessing the degree of forgetting. Formulas for these metrics are provided in Appendix D.

*Table 1.* Comparison of the proposed SEFE method with existing approaches, evaluated under TA metrics.

| Method | Accuracy on Each Task (%) | | | | | | | | Aggregate Results (%) | | | |
| --- | --- | --- | --- | --- | --- | --- | --- | --- | --- | --- | --- | --- |
| | SQA | $VQA^{Text}$ | ImgNet | GQA | VizWiz | Grd | $VQA^{v2}$ | $VQA^{OCR}$ | MFT↑ | MFN↑ | MAA↑ | BWT↑ |
| FFT | 2.95 | 36.38 | 52.35 | 46.40 | 33.90 | 0.00 | 61.65 | 50.00 | 65.87 | 35.45 | 36.73 | -30.42 |
| LoRA | 54.05 | 44.63 | 41.25 | 47.55 | 20.80 | 0.85 | 59.30 | 64.30 | **70.21** | 41.59 | 39.53 | -28.62 |
| O-LoRA | 75.40 | 52.89 | 71.85 | 47.30 | 37.35 | 7.10 | 61.85 | 61.20 | 69.30 | 51.87 | 49.56 | -17.43 |
| LoTA | 67.30 | 41.51 | 8.25 | 37.15 | 42.25 | 0.10 | 47.95 | 56.15 | 54.72 | 37.58 | 50.46 | -17.14 |
| FFT+ASD | 74.50 | 50.12 | 65.40 | 54.35 | 45.50 | 0.00 | 64.40 | 68.50 | 68.28 | 52.85 | 57.18 | -15.44 |
| LoRA+ASD | 74.45 | 49.70 | 39.30 | 52.00 | 50.45 | 7.05 | 62.25 | 47.80 | 68.13 | 47.88 | 59.71 | -20.26 |
| O-LoRA+ASD | 75.20 | 55.36 | 67.50 | 54.70 | 52.90 | 15.40 | 64.45 | 35.05 | 65.59 | 52.57 | 61.63 | -13.02 |
| LoTA+ASD | 76.90 | 42.65 | 15.85 | 40.25 | 45.10 | 0.30 | 54.35 | 54.00 | 56.99 | 41.18 | 56.28 | -15.82 |
| SEFE (Ours) | 75.35 | 58.66 | 83.10 | 54.25 | 48.85 | 16.75 | 65.35 | 66.25 | 69.02 | **58.57** | **63.04** | **-10.45** |

## 5.2. Implementation Details

In line with (Chen et al., 2024a), we utilize the pre-trained, instruction-untuned LLaVA-1.5 (Liu et al., 2024a) as the base model, which incorporates Vicuna-7B (Chiang et al., 2023) as its foundational LLM. Following the official configuration of LLaVA-1.5, the total batch size is set to 128, with learning rates of $2 \times 10^{-4}$ when using LoRA (Hu et al., 2022) and $2 \times 10^{-5}$ without it, and the LoRA rank parameter $r$ is set to 128. Other settings also remain consistent with those specified by LLaVA-1.5. For the ASD paradigm, we use the default value of 20 for the hyperparameter $X$, and data transformations requiring an MLLM utilize InternVL2-26B (OpenGVLab Team, 2024). For RegLoRA, we use default values of 2 for $M$ and $2.5 \times 10^3$ for $\lambda$. Our experiments are conducted on a node equipped with 8 NVIDIA A800 GPUs.

## 5.3. Comparison

To validate the effectiveness of our proposed SEFE, we compare it against existing approaches, including full-parameter fine-tuning (FFT), LoRA (Hu et al., 2022), O-LoRA (Wang et al., 2024b), and LoTA (Panda et al., 2024). Furthermore, to demonstrate that our ASD paradigm can mitigate *superficial forgetting* across various methods, we apply ASD to these existing approaches, *i.e.*, training them on CoIN-ASD.

The TA results presented in Table 1 show that: (1) ASD substantially enhances the performance of all tested methods, with average gains of 7.00%, 14.63%, and 7.27% in aggregate metrics MFN, MAA, and BWT, respectively. We attribute these improvements to ASD's ability to reduce *superficial forgetting*, which we further validate in subsequent experiments. Additionally, by bridging the answer-domain gap, ASD can reduce inter-task differences, which in turn lowers the model's update magnitude during new task learning. This mechanism may also indirectly alleviate *essential forgetting* to a slight degree. (2) Our SEFE approach outperforms all current state-of-the-art methods, even when ASD is applied to these methods. This highlights RegLoRA's ef-

*Table 2.* Evaluation of main components in our method.

| Configuration | Aggregate Results (%) | | | |
| --- | --- | --- | --- | --- |
| | MFT↑ | MFN↑ | MAA↑ | BWT↑ |
| Baseline (LoRA) | **70.21** | 41.59 | 39.53 | -28.62 |
| + ASD | 68.13 | 47.88 | 59.71 | -20.26 |
| + ASD + RegLoRA | 69.02 | **58.57** | **63.04** | **-10.45** |

fectiveness in addressing *essential forgetting* by preserving crucial prior knowledge, which works synergistically with ASD to achieve comprehensive forgetting reduction.

## 5.4. Ablation Study

In this section, we present ablation studies on SEFE, covering its main components, hyperparameter settings, and design choices. Due to space constraints, we only report aggregate metrics for these experiments.

### 5.4.1. MAIN COMPONENTS

Firstly, we evaluate the impact of the two main components: ASD and RegLoRA. Starting with LoRA as the baseline model, we incrementally incorporate each component. The quantitative results are shown in Table 2, and case studies are presented in Fig. 4.

Compared to the baseline, adding ASD results in significant improvements in three out of the four aggregate metrics: MFN increases by 6.29%, MAA by 20.18%, and BWT by 8.36%. We attribute these gains to a reduction in *superficial forgetting*, enabling the model to generate responses that reflect its true knowledge instead of being influenced by biased response patterns from recent tasks. This effect is illustrated in the case examples in Fig. 4. For instance, in the first two cases, the baseline model incorrectly provides the content of an option when answering a ScienceQA question (requiring only the option's letter), due to interference from the recently trained OCR-VQA task, which requires word or phrase responses. In case 3, after ImageNet training, the

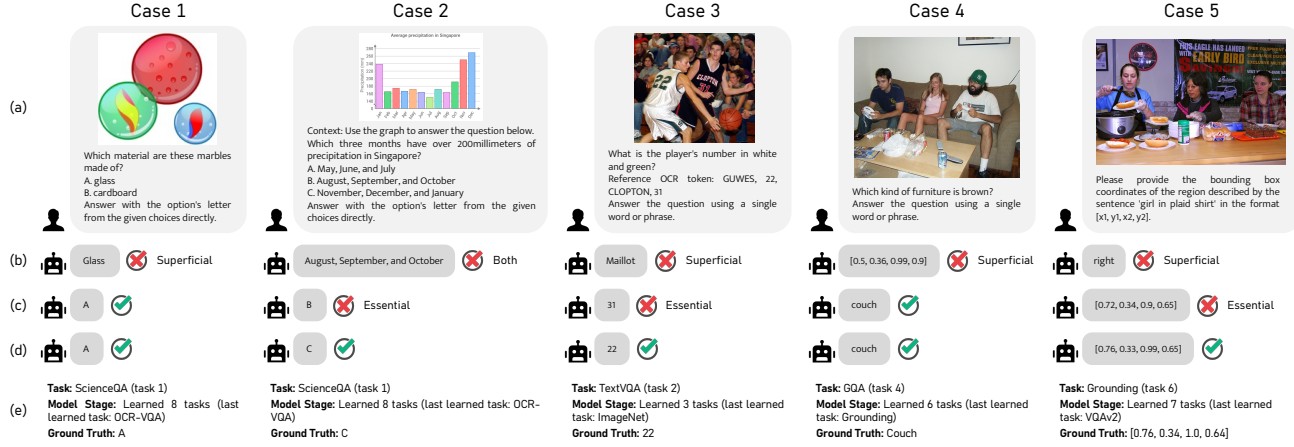

*Figure 4.* Case studies of main components in the proposed SEFE method. (a) Instruction; (b) Response from the baseline model (LoRA); (c) Response from the baseline model with ASD added; (d) Response from the baseline model with both ASD and RegLoRA added; (e) Basic information of the case. Additional case studies can be found in Appendix H.

baseline model mistakenly answers a TextVQA question with "Maillot", a category from ImageNet, instead of the required number. In case 4, after learning the Grounding task, the model incorrectly responds to a GQA task with a bounding box instead of a word. Similarly, in case 5, it responds to a Grounding question with a word instead of the required bounding box after learning the word-based VQAv2 task. These examples highlight the *superficial forgetting* observed in the baseline model. After integrating ASD, the model consistently produces responses that match the task requirements, even if the answers are not always entirely correct. This confirms ASD's effectiveness in mitigating *superficial forgetting* and facilitating a more accurate assessment of the model's knowledge.

Building on ASD, we introduce RegLoRA to address *essential forgetting*. As shown in Table 2, model performance further improves, with MFN, MAA, and BWT increasing by 10.69%, 3.33%, and 9.81%, respectively. This improvement indicates that RegLoRA effectively preserves the model's prior knowledge, allowing it to handle previously learned tasks more robustly. Cases 2 and 3 in Fig. 4 illustrate that the RegLoRA-enhanced model corrects the baseline model's erroneous answers. In case 5, the model's bounding box response improves its IoU with the GT from 0.49 to 0.90 after integrating RegLoRA, further confirming RegLoRA's role in enhancing response accuracy.

### 5.4.2. DATA TRANSFORMATION PROPORTION IN ASD

This section investigates the effect of varying the proportion of data transformations in ASD by adjusting the hyperparameter $X$. The results are presented in Table 3. The baseline condition ($X = 0$), where no transformation is applied, is shown in the first row. We observe that even a small trans-

*Table 3.* Comparison of data transformation proportions in ASD.

| $X$ | Aggregate Results (%) | | | |
|---|---|---|---|---|
| | MFT↑ | MFN↑ | MAA↑ | BWT↑ |
| 0 | **70.21** | 41.59 | 39.53 | -28.62 |
| 10 | 65.82 | 46.93 | 58.61 | -18.89 |
| 20 | 68.13 | **47.88** | **59.71** | -20.26 |
| 40 | 70.09 | 47.80 | 59.22 | -22.29 |
| 60 | 66.06 | 47.45 | 58.37 | -18.60 |
| 80 | 63.93 | 46.85 | 58.64 | **-17.08** |

*Table 4.* Comparison of regularized element proportions in RegLoRA.

| $M$ | Aggregate Results (%) | | | |
|---|---|---|---|---|
| | MFT↑ | MFN↑ | MAA↑ | BWT↑ |
| 0.5 | 67.18 | 55.76 | 61.53 | -11.43 |
| 1 | 68.62 | **58.71** | 62.38 | **-9.91** |
| 2 | **69.02** | 58.57 | **63.04** | -10.45 |
| 5 | 66.58 | 56.29 | 61.69 | -10.29 |
| 100 | 67.73 | 54.66 | 61.54 | -13.07 |

formation proportion of 10% significantly improves performance, indicating that introducing alternative response styles, even minimally, effectively reduces the model's bias toward a single style. Based on all four evaluation metrics, we select $X = 20$ as the default value, as it achieves optimal results in the key MFN and MAA metrics.

### 5.4.3. REGULARIZED ELEMENT PROPORTION IN REGLORA

In this section, we analyze the impact of varying the proportion of regularized elements, *i.e.*, the hyperparameter $M$,

Table 5. Comparison of regularization targets in RegLoRA.

| Regul. Tgt. | Aggregate Results (%) | | | |
|---|---|---|---|---|
| | MFT↑ | MFN↑ | MAA↑ | BWT↑ |
| $A$ | 68.43 | 47.20 | 59.49 | -21.23 |
| $B$ | 67.52 | 47.84 | 59.35 | -19.69 |
| $A$ & $B$ | 68.82 | 48.47 | 59.41 | -20.35 |
| $\Delta W$ (Ours) | **69.02** | **58.57** | **63.04** | **-10.45** |

on the performance of RegLoRA. The results are shown in Table 4. Our findings indicate that performance peaks when $M = 2$, suggesting that during LoRA fine-tuning, approximately 2% of updated parameters significantly influence task performance. When $M$ is set lower, the model struggles to retain knowledge from previous tasks, leading to a performance decrease. This insufficient knowledge retention may also limit initial performance on new tasks that closely relate to previous ones, which likely explains why MFT for $M = 0.5$ and $M = 1$ is lower than for $M = 2$. Conversely, setting $M$ too high overly restricts all parameter updates, impairing the model's ability to learn new information. Moreover, when regularization is applied to less critical elements, the energy allocated in the loss function for crucial elements decreases, adversely affecting the model's ability to retain prior knowledge. Hence, the performance for $M = 5$ and $M = 100$ is suboptimal. Based on these observations, we select $M = 2$ as the default setting.

5.4.4. REGULARIZATION TARGET IN REGLORA

In Table 5, we validate our decision to apply regularization to the top $M\%$ elements of the weight update matrix, $\Delta W$. The table compares this approach to the effects of regularizing the top $M\%$ elements in matrix $A$, matrix $B$, and in both matrices $A$ and $B$ together. The results show that regularizing $\Delta W$ yields the best performance. Since $\Delta W$ directly reflects model updates, whereas the other three options are indirect, regularizing $\Delta W$ is more effective in preserving prior knowledge. Additionally, because $\Delta W$ is not a model parameter, this choice leaves more capacity for parameter updates to accommodate new knowledge.

## 6. Conclusion

This paper provides an in-depth analysis of catastrophic forgetting in MCIT, categorizing it into *superficial forgetting* and *essential forgetting*. *Superficial forgetting* refers to the model's response style becoming biased by subsequent tasks, causing deviations from the required format of previous tasks. On the other hand, *essential forgetting* indicates the loss of knowledge, resulting in factual errors in responses. To address these issues, we propose the SEFE method, which introduces two components: the ASD paradigm and RegLoRA. ASD mitigates *superficial*

*forgetting* by diversifying question types within a single task, thereby preventing response style bias, enhancing performance, and enabling reliable assessment of knowledge state. Using ASD, we processed the CoIN benchmark to create the new CoIN-ASD benchmark, allowing future methods to evaluate knowledge retention without interference from response style biases. To address *essential forgetting*, RegLoRA identifies critical elements in the weight update matrix of past LoRAs and applies a regularization loss to minimize corresponding elements in future LoRAs. Experimental results demonstrate the effectiveness of ASD and RegLoRA, as well as the state-of-the-art performance of SEFE as a comprehensive solution.

## Acknowledgements

This work was supported in part by the National Natural Science Foundation of China under Grants 62471278 and 62306220, in part by the Research Grants Council of the Hong Kong Special Administrative Region, China under Grant STG5/E-103/24-R, and in part by the Taishan Scholar Project of Shandong Province under Grant tsqn202306079.

## Impact Statement

This paper presents work whose goal is to advance the field of Machine Learning. There are many potential societal consequences of our work, none which we feel must be specifically highlighted here.

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

# A. Additional Related Work

## A.1. Multimodal Large Language Model

Multimodal Large Language Models (MLLMs) have become one of the most prominent research areas today. GPT-4/4o (Achiam et al., 2023) is the most representative work in this area, demonstrating exceptional performance across a wide range of multimodal tasks. Subsequent models, such as Gemini (Gemini Team, 2023) and Claude (Anthropic, 2024), also showcase impressive performance, offering user experiences that rival or even exceed those of GPT-4/4o.

In the open-source community, several noteworthy models have also emerged. For example, InstructBLIP (Dai et al., 2023) builds upon the BLIP2 (Li et al., 2023b) pretraining model and incorporates instruction tuning to improve the model's ability to understand user commands and process text-image queries. LLaVA (Liu et al., 2023) adopts a simple yet efficient architecture and introduces an effective method for generating visual instruction tuning data using large language models (LLMs), achieving strong performance. LLaVA-1.5 (Liu et al., 2024a), an enhanced version of LLaVA, refines design choices such as the vision-language connector, resulting in further improvements in performance. QwenVL (Bai et al., 2023b) proposes a three-stage training paradigm, using images with varying resolutions at different stages of training, optimizing both image detail comprehension and training efficiency. InternVL (Chen et al., 2024d) focuses on scaling up the visual encoder's parameter size and employs a different three-stage training strategy to align the large visual encoder with the text encoder, enhancing the model's visual understanding capabilities.

## A.2. LoRA Variants

In addition to the previously discussed O-LoRA (Wang et al., 2024b), several LoRA variants have been proposed from perspectives beyond mitigating catastrophic forgetting. For example, AdaLoRA (Zhang et al., 2023) adaptively allocates the parameter budget across weight matrices based on their importance scores. LoRA+ (Hayou et al., 2024) assigns different learning rates to matrices $A$ and $B$ to improve performance in settings with large embedding dimensions. DoRA (Liu et al., 2024b) decomposes pretrained weights into magnitude and direction components, applying LoRA solely to the direction component to reduce the number of trainable parameters. Imbalance-Regularized LoRA (Zhu et al., 2024b), which shares a similar name with our proposed RegLoRA, introduces a regularization term $\left| AA^\top - \frac{r}{k} B^\top B \right|_F^2$ to improve stability during the forward pass and enhance downstream task performance. Overall, these approaches are fundamentally different from our RegLoRA in terms of methodology and objectives.

## A.3. Understanding Catastrophic Forgetting

Several existing works also aim to understand the phenomenon of catastrophic forgetting. For instance, (Li et al., 2024c) shows that catastrophic forgetting during LLM fine-tuning becomes more pronounced as the loss landscape sharpens, suggesting a strong positive correlation between sharpness and forgetting. (Zhai et al., 2023) argues that in MLLMs, catastrophic forgetting arises as fine-tuning shifts the model's focus from general visual-text alignment to dataset-specific overfitting, resulting in performance degradation even when the vision encoder is frozen. Among these studies, (Zheng et al., 2025) is most relevant to ours. It introduces the concept of *spurious forgetting*, where the model loses task alignment without any genuine loss of knowledge. This notion is partially similar with our definition of *superficial forgetting*. However, *spurious forgetting* emphasizes the recoverability and assumes that no actual knowledge has been forgotten. In contrast, *superficial forgetting* does not make assumptions about recoverability or knowledge retention.

# B. Details of ASD

## B.1. Transformation Rules

Table 6 details the transformation rules in the ASD paradigm, providing guidelines for converting various source formats into specific target formats: yes/no questions (Y/N), multiple-choice questions (MCQ), short answer questions (short), brief explanation/description questions (brief), and detailed explanation/description questions (detail). These rules specify the Response Format Prompt (RFP) and methods for rewriting both the instruction and ground-truth (GT) labels. The outlined transformations serve as general guidelines for typical scenarios but may require minor adjustments for specific applications. Fig. 5 illustrates several transformation examples, demonstrating the conversion of five original question types into the other four formats, thereby offering a clearer understanding of the transformation process in ASD.

*Table 6.* Specific Transformation Rules of ASD.

| Target | Source | RFP | Instruction | GT Label |
|---|---|---|---|---|
| Y/N | MCQ | Is the answer correct? Answer 'Yes' or 'No'. | Structured as "question + a potentially correct answer + RFP". The question is the original question. For half of the samples, the potentially correct answer is the correct answer (*i.e.*, the original GT label); for the other half, it is an incorrect answer. — When an incorrect answer is required, randomly select one of the incorrect options. | Use "Yes" to samples where the potentially correct answer is correct; use "No" to the others. |
| | Short | | When an incorrect answer is required, generate a plausible but incorrect distractor using MLLMs. | |
| | Brief | | | |
| | Detail | | | |
| MCQ | Y/N | Answer with the option's letter from the given choices directly. | Structured as "question + option list + RFP". The question is the original question, and an option list needs to be constructed. — The option list includes two options: "A. Yes" and "B. No". | Use the letter corresponding to the correct option. |
| | Short | | The option list include four randomly ordered options, one being the original GT label and the remaining three being plausible but incorrect distractors generated by MLLMs. | |
| | Brief | | | |
| | Detail | | | |
| Short | Y/N | Answer the question using a single word or phrase. | Structured as "question + RFP". — For half of the samples, the question is the original question; for the other half, rewrite the original question by MLLMs into a similarly focused questions that require answers other than "Yes" or "No". | When the question is the original question, use the original "Yes" or "No"; When the question is rewritten by MLLMs, use the answer generated together with the question. |
| | MCQ | | The question is the original question. | Use the content of the correct option. |
| | Brief | | | Shorten the original GT label to within 10 words by MLLMs. |
| | Detail | | | |
| Brief | Y/N | Answer the question and provide a brief explanation. | Structured as "question + RFP". | Structured as "direct answer + explanation". The direct answer is the original GT label; the explanation is about 20 words, generated by MLLMs. |
| | MCQ | | | |
| | Short | | | |
| | Detail | Answer the question using a brief explanation/description | | Adjust the original GT label to approximately 20 words by MLLMs. |
| Detail | Y/N | Answer the question and provide a detailed explanation. | | Structured as "direct answer + explanation". The direct answer is the original GT label; the explanation is about 50 words, generated by MLLMs. |
| | MCQ | | | |
| | Short | | | |
| | Brief | Answer the question using a detailed explanation/description | | Adjust the original GT label to approximately 50 words by MLLMs. |

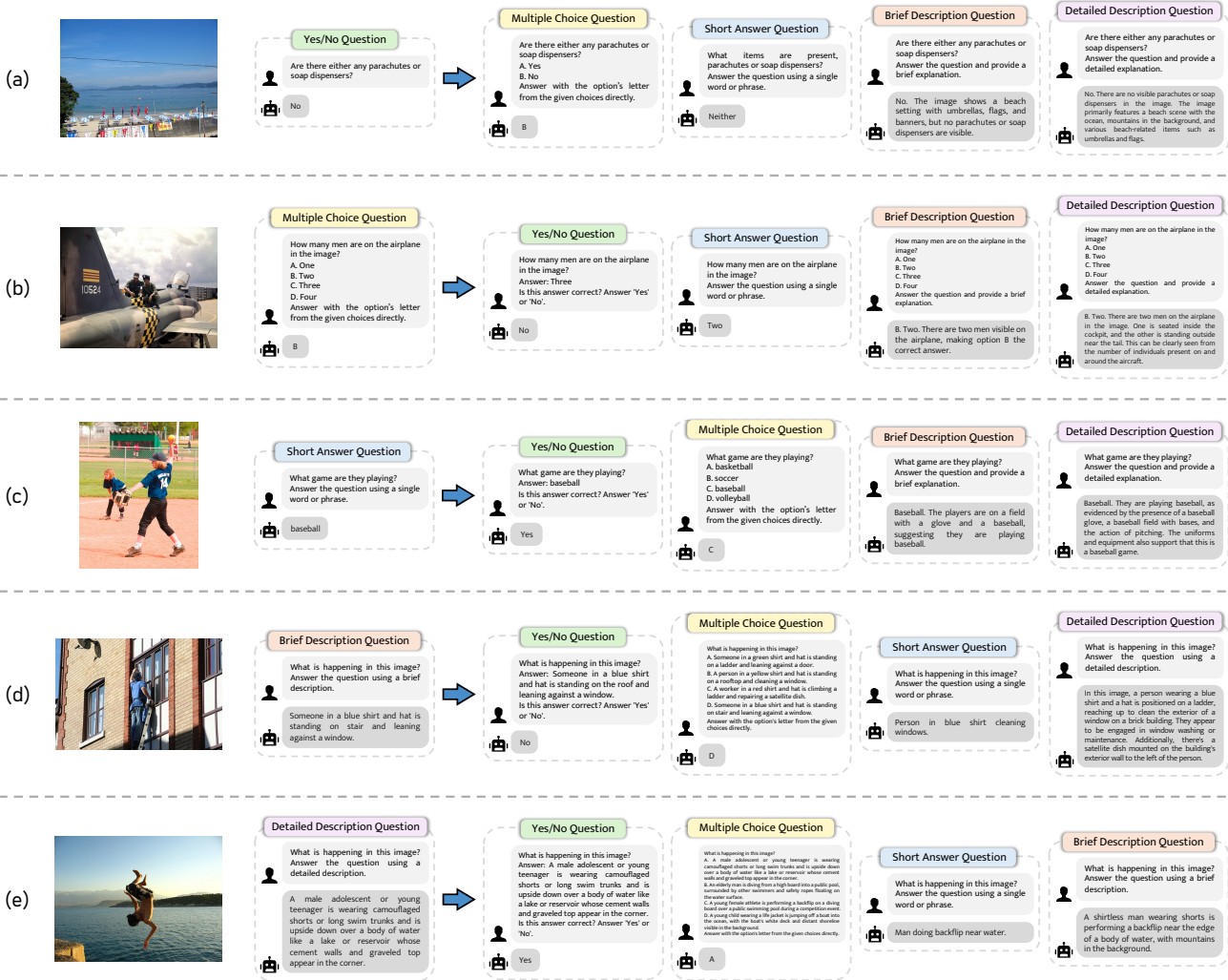

*Figure 5.* Illustrative transformations of the ASD paradigm: (a) Conversion of a yes/no question into four other question types; (b) Conversion of a multiple-choice question into four other question types; (c) Conversion of a short-answer question into four other question types; (d) Conversion of a brief description question into four other question types; (e) Conversion of a detailed description question into four other question types.

### B.2. Transformation Details of CoIN-ASD

The CoIN-ASD benchmark is a modified version of the CoIN benchmark (Chen et al., 2024a), tailored using the ASD paradigm to assess *essential forgetting*. Like CoIN, CoIN-ASD comprises eight tasks in sequence: ScienceQA (Lu et al., 2022), TextVQA (Singh et al., 2019), ImageNet (Deng et al., 2009), GQA (Hudson & Manning, 2019), VizWiz (Gurari et al., 2018), Grounding (Kazemzadeh et al., 2014; Mao et al., 2016), VQAv2 (Goyal et al., 2017), and OCR-VQA (Mishra et al., 2019). In constructing the CoIN-ASD dataset, we primarily adhere to the transformation rules outlined in Sec. B.1, with minor modifications in three specific instances:

1. **ScienceQA questions to short answer format:** ScienceQA questions (Lu et al., 2022) are initially MCQs, where some samples require the provided options to determine the correct answer. For example, questions such as *"Which state among the options is the northernmost?"* require explicit options to identify the correct response. To accommodate this, we include a reformatted option list when converting ScienceQA questions into short answer ones, structured as *"[Option 1], [Option 2], . . . , or [Option N]"*. Additionally, the RFP is adjusted to a more natural one: *"Directly give the answer."*

2. **ImageNet and Grounding questions to Y/N or MCQ:** In these cases, employing an MLLM for distractor generation is unnecessary. For ImageNet questions, we randomly select distractors from the class list. For Grounding questions, we randomly generate bounding boxes with an Intersection-over-Union (IoU) less than 0.5 compared to the GT bounding box. For other tasks with similarly constrained answer spaces, reformulating them as Y/N or MCQ can also eliminate the need for MLLM-based distractor generation using similar strategies.

3. **Grounding questions to brief or detailed format:** Instead of using a response format that combines a direct answer with an explanation, we reformulate the questions to request descriptions of the content within the given bounding box, which is more natural.

All other transformations strictly follow the rules defined in Sec. B.1. Further details on the prompts and transformed data examples are available in our code repository.

## C. Loss Function

The overall loss function of the model is defined as:

$$\mathcal{L}_{total} = \mathcal{L}_{lm} + \mathcal{L}_{reg}, \tag{3}$$

where $\mathcal{L}_{reg}$ is the regularization loss defined in Eq. 2, and $\mathcal{L}_{lm}$ is the standard language modeling loss, *i.e.*, the next-token prediction loss, given by:

$$\mathcal{L}_{lm} = -\sum_{i=1}^{L} \log P(x_i | X_v, X_{instruct,<i}, X_{a,<i}), \tag{4}$$

where $L$ denotes the length of the answer, $X_v$ represents the visual input tokens, and $X_{instruct,<i}$ and $X_{a,<i}$ denote the instruction and answer tokens preceding the current token $x_i$, respectively.

Specifically, during training on the first task, there is no prior knowledge to preserve, so the objective reduces to the language modeling loss alone. For all subsequent tasks, the objective corresponds to the complete loss defined in Eq. 3.

## D. Definitions and Formulas of Metrics

In this section, we present the definitions and mathematical expressions of the four aggregate metrics employed in our evaluation: Mean Fine-tune Accuracy (MFT), Mean Final Accuracy (MFN), Mean Average Accuracy (MAA), and Backward Transfer (BWT).

MFT represents the average accuracy of each task immediately after it is learned. This metric serves as an upper-bound performance indicator, indicating the model's performance without any forgetting. It is calculated as:

$$MFT = \frac{1}{T} \sum_{i=1}^{T} A_{i,i}, \tag{5}$$

*Table 7.* Comparison of the proposed SEFE method with existing approaches, evaluated under KC metrics.

| Method | Accuracy on Each Task (%) | | | | | | | | Aggregate Results (%) | | | |
|---|---|---|---|---|---|---|---|---|---|---|---|---|
| | *SQA* | *VQA$^{Text}$* | *ImgNet* | *GQA* | *VizWiz* | *Grd* | *VQA$^{v2}$* | *VQA$^{OCR}$* | MFT↑ | MFN↑ | MAA↑ | BWT↑ |
| FFT | 73.46 | 57.61 | 81.79 | 59.30 | 45.97 | 77.75 | 71.84 | 70.64 | 75.94 | 67.29 | 68.06 | -8.65 |
| LoRA | 74.11 | 62.59 | 64.21 | 61.50 | 38.14 | 43.32 | 69.64 | 80.47 | **78.49** | 61.75 | 68.34 | -16.74 |
| O-LoRA | 84.45 | 67.50 | 84.67 | 60.40 | 52.41 | 60.85 | 70.97 | 78.28 | 77.86 | 69.94 | 71.87 | -7.91 |
| LoTA | 76.54 | 62.93 | 33.28 | 54.31 | 57.12 | 19.31 | 61.73 | 76.30 | 67.84 | 55.19 | 64.42 | -12.66 |
| FFT+ASD | 80.25 | 67.57 | 81.88 | 66.09 | 59.73 | 9.10 | 74.43 | 85.04 | 76.17 | 65.51 | 71.44 | -10.66 |
| LoRA+ASD | 80.09 | 66.13 | 72.39 | 64.46 | 66.89 | 59.40 | 72.06 | 75.29 | 77.29 | 69.59 | 73.36 | -7.71 |
| O-LoRA+ASD | 80.82 | 69.43 | 86.37 | 66.87 | 67.33 | 59.34 | 73.59 | 67.43 | 75.94 | 71.40 | 75.07 | **-4.54** |
| LoTA+ASD | 82.49 | 63.01 | 44.75 | 56.99 | 58.42 | 36.98 | 67.53 | 76.42 | 70.38 | 60.82 | 70.12 | -9.56 |
| SEFE (Ours) | 80.79 | 70.59 | 89.05 | 66.44 | 63.81 | 57.90 | 74.41 | 83.36 | 77.98 | **73.29** | **75.61** | -4.69 |

*Table 8.* Evaluation of main components in our method, evaluated under KC metrics.

| Configuration | Aggregate Results (%) | | | |
|---|---|---|---|---|
| | MFT↑ | MFN↑ | MAA↑ | BWT↑ |
| Baseline (LoRA) | **78.49** | 61.75 | 68.34 | -16.74 |
| + ASD | 77.29 | 69.59 | 73.36 | -7.71 |
| + ASD + RegLoRA | 77.98 | **73.29** | **75.61** | **-4.69** |

where $T$ is the total number of tasks, and $A_{i,i}$ denotes the accuracy on task $i$ immediately after learning it.

MFN measures the average accuracy across all tasks after the model has completed the entire incremental training sequence. This metric reflects the model's final performance after all tasks have been learned. It is defined as:

$$MFN = \frac{1}{T} \sum_{i=1}^{T} A_{T,i}, \qquad (6)$$

where $A_{T,i}$ represents the accuracy on task $i$ after learning all $T$ tasks.

MAA provides a comprehensive measure of the model's performance throughout the entire training process. It is the mean of the average accuracies on all learned tasks after each incremental training step. The formula for MAA is:

$$MAA = \frac{1}{T} \sum_{j=1}^{T} (\frac{1}{j} \sum_{i=1}^{j} A_{j,i}), \qquad (7)$$

where $A_{j,i}$ denotes the accuracy on task $i$ after the model has been trained on the first $j$ tasks.

BWT assesses the extent of forgetting by measuring the difference in accuracy for each task between the final training step and immediately after the task was learned. It is calculated as:

$$BWT = \frac{1}{T} \sum_{i=1}^{T} (A_{T,i} - A_{i,i}). \qquad (8)$$

A negative BWT value indicates forgetting, with larger negative values implying greater forgetting.

# E. Knowledge Capability

In this section, we present the evaluation results based on the Knowledge Capability (KC) metrics. As defined in the CoIN benchmark (Chen et al., 2024a), KC metrics involve utilizing Qwen1.5-32B (Bai et al., 2023a) to assess the knowledge reflected in model responses.

Table 7 compares our method with existing approaches using KC metrics. Additionally, it includes results incorporating the ASD paradigm into existing methods. This table parallels the Truth Alignment (TA) metrics reported in Table 1. The

*Table 9.* Comparison of data transformation proportions in ASD, evaluated under KC metrics.

| $X$ | Aggregate Results (%) | | | |
|---|---|---|---|---|
| | MFT↑ | MFN↑ | MAA↑ | BWT↑ |
| 0 | **78.49** | 61.75 | 68.34 | -16.74 |
| 10 | 75.62 | 68.18 | 72.24 | -7.44 |
| 20 | 77.29 | **69.59** | **73.36** | -7.71 |
| 40 | 78.30 | 66.85 | 72.41 | -11.45 |
| 60 | 75.63 | 67.77 | 71.97 | -7.86 |
| 80 | 74.17 | 67.59 | 72.02 | **-6.58** |

*Table 10.* Comparison of regularized element proportions in RegLoRA, evaluated under KC metrics.

| $M$ | Aggregate Results (%) | | | |
|---|---|---|---|---|
| | MFT↑ | MFN↑ | MAA↑ | BWT↑ |
| 0.5 | 76.89 | 73.26 | 74.79 | **-3.63** |
| 1 | 77.74 | **74.06** | 75.29 | -3.68 |
| 2 | **77.98** | 73.29 | **75.61** | -4.69 |
| 5 | 76.79 | 73.09 | 74.90 | -3.70 |
| 100 | 77.41 | 72.09 | 74.80 | -5.32 |

observed trends in KC metrics align closely with those in TA metrics. First, integrating ASD improves performance across various methods. Although KC metrics primarily assess knowledge rather than instruction-following—an aspect theoretically unaffected by *superficial forgetting*—severe *superficial forgetting* can obscure a model's true knowledge capabilities (as illustrated in Fig. 1 and Fig. 4). This explains the enhancement in KC metrics achieved with ASD. Second, even with ASD integration, existing methods still underperform compared to our SEFE approach, further demonstrating the effectiveness of RegLoRA in mitigating *essential forgetting*.

Table 8 evaluates the main components of our method. Table 9 explores the impact of varying data transformation proportions in ASD, *i.e.*, hyperparameter $X$. Table 10 investigates the effect of different proportions of regularization elements in RegLoRA, *i.e.*, hyperparameter $M$. Finally, Table 11 examines the influence of different regularization targets in RegLoRA. These results correspond to the TA results presented in Tables 2, 3, 4, and 5 of the main text, and similar trends are observed in both KC and TA metrics. Specifically, Table 8 confirms the substantial advantage of combining both core components. Table 9 identifies 20% data transformation in ASD as optimal. Finally, Tables 10 and 11 demonstrate that focusing regularization on the top 2% of elements in the weight update matrix $\Delta W$ yields the best performance in RegLoRA.

While the KC results generally align with the trends observed in the TA results and appear reasonable overall, certain failure cases were identified during evaluation. For example, when a model's response is "Yes", Qwen1.5-32B (Bai et al., 2023a) often assigns a high score to it, even when this answer is highly unreliable or entirely irrelevant to the question. These issues may occur at a similar rate across different experimental settings, potentially mitigating their impact on the overall trends. Nevertheless, due to these limitations, we consider the TA results to be the most reliable, with the KC results serving as a supplementary reference.

*Table 11.* Comparison of regularization targets in RegLoRA, evaluated under KC metrics.

| Regul. Tgt. | Aggregate Results (%) | | | |
|---|---|---|---|---|
| | MFT↑ | MFN↑ | MAA↑ | BWT↑ |
| $A$ | 77.84 | 68.83 | 73.30 | -9.01 |
| $B$ | 77.33 | 69.55 | 73.09 | -7.78 |
| $A$ & $B$ | 77.75 | 69.32 | 73.08 | -8.43 |
| $\Delta W$ (Ours) | **77.98** | **73.29** | **75.61** | **-4.69** |

*Table 12.* Comparison of different MLLMs for data transformation in ASD, evaluated under TA and KC metrics. "N/A" denotes the absence of ASD.

| MLLM | TA Aggregate Results (%) | | | | KC Aggregate Results (%) | | | |
|---|---|---|---|---|---|---|---|---|
| | MFT↑ | MFN↑ | MAA↑ | BWT↑ | MFT↑ | MFN↑ | MAA↑ | BWT↑ |
| N/A | **70.21** | 41.59 | 39.53 | -28.62 | **78.49** | 61.75 | 68.34 | -16.74 |
| InternVL2-8B | 66.80 | **48.72** | 59.17 | **-18.08** | 76.28 | 69.21 | 72.73 | **-7.06** |
| InternVL2-26B | 68.13 | 47.88 | **59.71** | -20.26 | 77.29 | **69.59** | **73.36** | -7.71 |

*Table 13.* Comparison of different $\lambda$ values for RegLoRA, evaluated under TA and KC metrics.

| $\lambda$ | TA Aggregate Results (%) | | | | KC Aggregate Results (%) | | | |
|---|---|---|---|---|---|---|---|---|
| | MFT↑ | MFN↑ | MAA↑ | BWT↑ | MFT↑ | MFN↑ | MAA↑ | BWT↑ |
| $1 \times 10^3$ | **69.83** | 56.50 | 61.79 | -13.33 | **78.39** | 73.28 | 74.74 | -5.11 |
| $2.5 \times 10^3$ | 69.02 | **58.57** | **63.04** | -10.45 | 77.98 | **73.29** | **75.61** | -4.69 |
| $5 \times 10^3$ | 68.17 | 58.52 | 62.54 | **-9.65** | 77.40 | 73.09 | 75.25 | **-4.30** |

## F. Different MLLMs in ASD

To evaluate the impact of data quality generated by MLLMs on ASD, we compare the performance of models trained with data produced by InternVL2-26B (our default) and InternVL2-8B (OpenGVLab Team, 2024). The results are presented in Table 12. Since both MLLMs belong to the InternVL2 series, this comparison controls for additional confounding factors, suggesting that larger models produce higher-quality data. The results show that the model trained with InternVL2-26B data performs slightly better on the most critical MAA metric. However, overall, there is no significant difference compared to the model trained with InternVL2-8B data. Regardless of the MLLM used, both models significantly outperform the baseline without ASD (denoted as "N/A" in the first row). These findings suggest that the effectiveness of ASD may not strongly depend on the quality of MLLM-generated data. Instead, ASD's primary advantage lies in exposing the model to diverse answering styles during task learning, thereby mitigating biases toward any single style.

## G. Hyperparameter $\lambda$ in RegLoRA

Table 13 presents the aggregate metrics for TA and KC under varying values of the regularization loss weight hyperparameter $\lambda$ in RegLoRA. As shown, smaller $\lambda$ values lead to higher MFT scores, indicating improved acquisition of new knowledge. In contrast, larger $\lambda$ values result in higher BWT scores, signifying reduced forgetting of previously learned knowledge. Overall, the findings suggest that $\lambda = 2.5 \times 10^3$ achieves the optimal trade-off between these objectives, yielding the highest performance in the key metrics, MFN and MAA, for both TA and KC.

## H. Additional Case Studies

In Fig. 6, we present additional case studies to complement those discussed in Fig. 4 of the main text. Specifically, cases 1–3 involve ScienceQA scenarios (Lu et al., 2022) where the expected answers are the letters of correct options. However, the baseline model, after learning the ImageNet task (Deng et al., 2009), incorrectly outputs the ImageNet category "Cucumber". Similarly, after learning the word-based VQAv2 task (Goyal et al., 2017), the model erroneously outputs the content of the options. By incorporating ASD, the model produces answers in the correct format, and further adding RegLoRA corrects errors in some cases.

Cases 4–6 require responses of specified ImageNet categories as defined by the CoIN benchmark (Chen et al., 2024a). However, after learning word-based tasks such as GQA (Hudson & Manning, 2019) or VizWiz (Gurari et al., 2018), the baseline model fails to provide valid ImageNet categories. In contrast, applying ASD ensures the model generates valid categories for all examples, with RegLoRA further enhancing prediction accuracy.

Cases 7–8 involve short answer questions, where the expected outputs are words or phrases. The baseline model, influenced by recent training on the Grounding task (Kazemzadeh et al., 2014; Mao et al., 2016), instead outputs bounding boxes, making it difficult to assess whether relevant knowledge is retained. Similar to previous cases, incorporating ASD corrects the output format, while RegLoRA further enhances the accuracy of the responses.

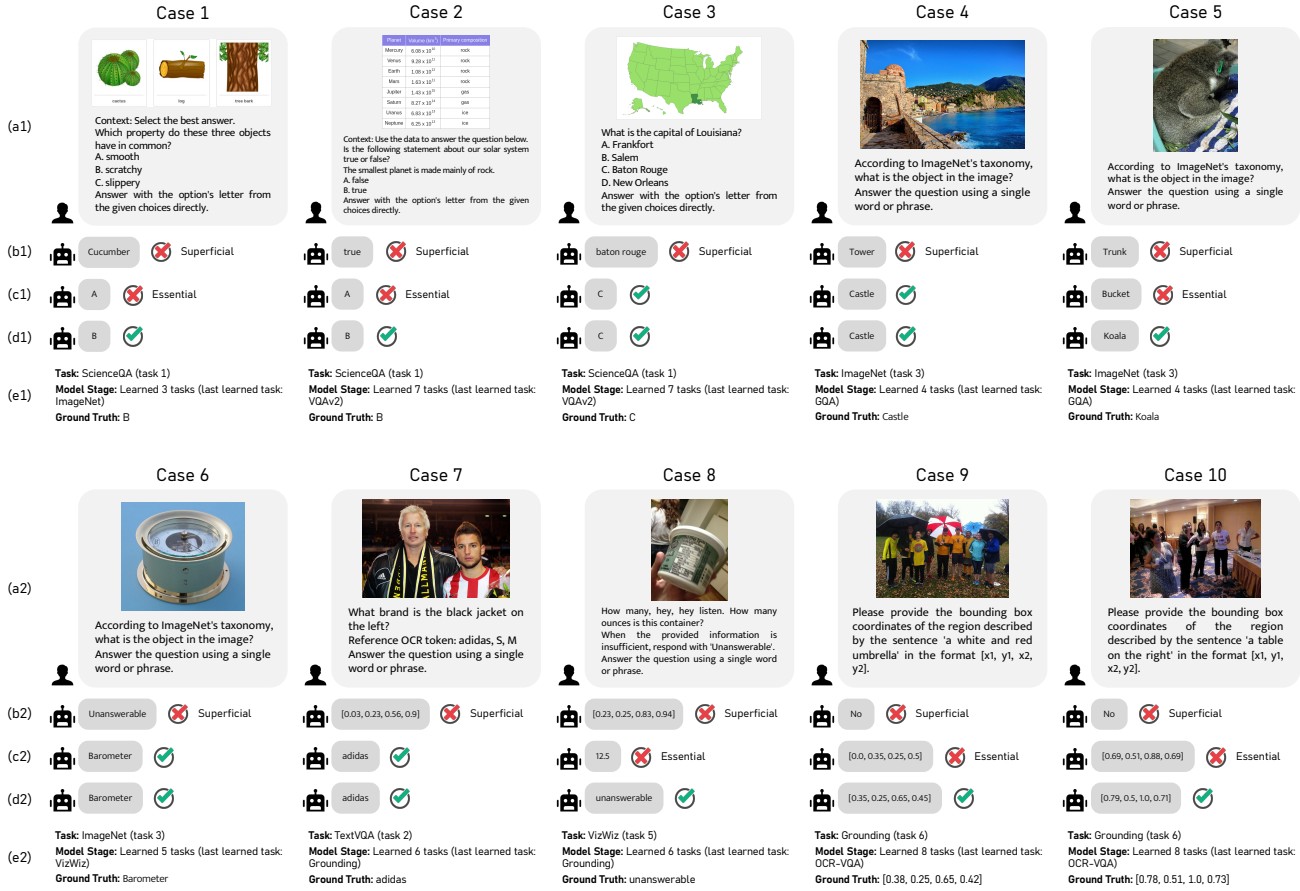

*Figure 6.* Additional case studies of main components in the proposed SEFE method. (a) Instruction; (b) Response from the baseline model (LoRA); (c) Response from the baseline model with ASD added; (d) Response from the baseline model with both ASD and RegLoRA added; (e) Basic information of the case.

Finally, cases 9–10 pertain to the Grounding task. After exposure to the word-based OCR-VQA task ([Mishra et al., 2019](#)), the baseline model generates words instead of the required bounding boxes. After applying ASD, the model generates responses in the correct format, and the addition of RegLoRA significantly improves the IoU between the predicted bounding boxes and the GT. Specifically, the IoU for case 9 increases from 0.0 to 0.76, and for case 10, it rises from 0.28 to 0.83. These results reaffirm the efficacy of our ASD method in mitigating *superficial forgetting* and the role of RegLoRA in addressing *essential forgetting*.

