# OpenReview forum: "SEFE: Superficial and Essential Forgetting Eliminator for Multimodal Continual Instruction Tuning"
_ICML.cc/2025/Conference — ICML 2025 poster_

### Official Review · Reviewer_HyXf · 2025-03-06

**Overall Recommendation:** 4

**Summary:**

This paper focuses on the continual learning task for multimodal large models, termed MCIT. The authors innovatively categorize catastrophic forgetting issues in this field into two distinct types: superficial and essential forgetting. They propose the SEFE model, which incorporates two key modules, ASD and RegLoRA, to respectively address these two categories of forgetting problems. Experimental results demonstrate the effectiveness of the proposed model. Furthermore, the authors introduce a novel dataset, CoIN-ASD, specifically designed to evaluate the performance of MCIT tasks. The article exhibits a well-organized structure and maintains clear semantic coherence throughout its presentation.

## Update After Rebuttal
I thank the authors' rebuttal and keep my rating to accept this paper.

**Claims And Evidence:**

The claims are generally well-supported by experimental results, demonstrating the effectiveness of SEFE, ASD, and RegLoRA.

**Essential References Not Discussed:**

The authors have provided the relevant references to support their work.

**Experimental Designs Or Analyses:**

The experimental design is well-structured and effectively evaluates the proposed methods for addressing catastrophic forgetting. The distinction between superficial and essential forgetting is clearly tested, and the results demonstrate the effectiveness of ASD and RegLoRA.

**Methods And Evaluation Criteria:**

The proposed methods and evaluation criteria are well-structured and align with the challenges of incremental learning in MLLMs. The distinction between superficial and essential forgetting is insightful, and ASD and RegLoRA offer effective solutions. While the approach is promising, providing clearer explanations in certain areas would further strengthen the presentation and impact of the work.

**Other Comments Or Suggestions:**

1. Is there a difference between RegLoRA and the regularization-based methods in continual learning? Both seem to constrain the update of some parameters during training to adapt to new tasks.

2.Why does ASD convert tasks into these five ways? Is there any prior knowledge? Or are these five ways sufficient to answer all questions in the real world?

3.In CoIN-ASD, why only a part of the questions are converted into other forms? Would it be better if all questions were converted according to the five forms?

4.Does it need to train a RegLoRA for each new task? How are the RegLoRAs of different tasks combined? What are the overall input, output, training, and testing processes of the model? Can a schematic diagram or pseudo-code be provided?

**Other Strengths And Weaknesses:**

Strengths:

1.The author divides the catastrophic forgetting problem in MCIT into superficial and essential forgetting problems, and proposes new methods, ASD and RegLoRA, to address them.

2.The author proves the effectiveness of the proposed method through experiments.

3.The author puts forward a new dataset for the MCIT task.

4.The overall structure of the article is clear and easy to understand.

Weaknesses:

1.The abstract is too long and needs to be shortened.

2.The motivation and implementation of the module design are not clearly explained.

**Questions For Authors:**

No

**Relation To Broader Scientific Literature:**

The authors are the first to break down the issue of catastrophic forgetting into two categories: superficial and essential forgetting problems. They propose corresponding methods to address each, providing new insights and solutions for the development of this field. This distinction offers a fresh perspective on understanding and mitigating catastrophic forgetting, contributing significantly to advancing current research.

**Theoretical Claims:**

For catastrophic forgetting, the authors address the problem in two parts, distinguishing between superficial and essential forgetting. The proposed methods, ASD and RegLoRA, effectively mitigate these issues, providing a well-reasoned and empirically supported solution. While the paper does not focus on formal theoretical proofs, the conceptual framework is sound and aligns well with the task objectives.

---

> ### Author Rebuttal · Authors · 2025-03-29
>
> Thank you for recognizing the strengths of our paper and for your effort in reviewing our submission. Below is a detailed response to your concerns:
>
> ### **The Abstract Should Be Shortened**
> Thank you for the suggestion. We will shorten the abstract in the revised version to improve conciseness.
>
> ### **Difference Between RegLoRA and Existing Regularization-Based Methods**
> RegLoRA can be viewed as a regularization-based method. However, it differs from most existing approaches in several key aspects:
>
> - As LoRA fine-tuning becomes increasingly prevalent for adapting large-scale models, there is a growing need to mitigate catastrophic forgetting in LoRA-based settings. In this context, RegLoRA—which applies regularization to LoRA adapters—is naturally well-suited, whereas existing regularization methods that operate on the full model may be less appropriate.
> - Unlike some existing regularization-based approaches, RegLoRA does not require auxiliary models or a significant number of additional parameters on GPUs. This substantially reduces GPU memory usage.
> - RegLoRA focuses on regularizing a small number of key elements rather than entire weight matrices or feature maps, which also distinguishes it from most existing methods in terms of strategy.
>
> In summary, while RegLoRA belongs to the family of regularization-based methods, its design choices make it more efficient and practical for our task setting.
>
> ### **Rationale for Converting Tasks in Five Ways**
> Our Answer Style Diversification (ASD) paradigm reformulates tasks into five question types based on prior knowledge. Specifically, we examined 15 widely-used MLLM benchmarks and analyzed the applications of MLLM in real-world scenarios. Our analysis reveals that the identified categories—yes/no questions, multiple-choice questions, short answer questions, brief explanation/description questions, and detailed explanation/description questions—cover the majority of use cases. The list of benchmarks referenced is provided in lines L97–L101, all of which belong to these five types. In practical settings, if additional question types arise in specific domains, our approach can also be extended to accommodate them.
>
> ### **Reason for Partially Converting Questions**
> If all questions are converted, the model cannot learn the answer format of the original question style. Since the test set retains the original format, a model that has never seen samples in this style is unlikely to perform well during evaluation. As shown in Table 3, when the conversion ratio is high (*e.g.*, 80%), performance drops compared to lower ratios (*e.g.*, 20%), indicating that preserving a substantial portion of original-format questions is essential.
>
> ### **Possibility to Combining RegLoRAs and Pseudo-Code Explanation**
> We agree that combining RegLoRAs—or more generally, LoRAs—from different tasks is an intriguing and promising direction. If successful, this approach could enable the construction of a unified multi-faceted MLLM by integrating task-specific LoRAs, which holds significant practical value. However, this idea lies beyond the current scope of our work. While we are not yet able to evaluate its feasibility or propose a concrete method, we appreciate your insightful suggestion and will continue exploring this direction in future research.
>
> Regarding the overall training procedure of RegLoRA, we provide the pseudo-code below for clarity:
>
> ---
>
> **Require:** base MLLM $M_0$, training sets of all tasks {$D_1, D_2,...,D_T$}
> **for** task $j$ in $1$ to $T$ **do**
> &nbsp;&nbsp;&nbsp;&nbsp;Insert a new LoRA adapter $LoRA_j$ into model $M_{j-1}$
> &nbsp;&nbsp;&nbsp;&nbsp;**for** each batch in $D_j$ **do**
> &nbsp;&nbsp;&nbsp;&nbsp;&nbsp;&nbsp;&nbsp;&nbsp;Compute language modeling loss $L_{lm}$
> &nbsp;&nbsp;&nbsp;&nbsp;&nbsp;&nbsp;&nbsp;&nbsp;**if** j > 1 **then**
> &nbsp;&nbsp;&nbsp;&nbsp;&nbsp;&nbsp;&nbsp;&nbsp;&nbsp;&nbsp;&nbsp;&nbsp;Compute regularization loss $L_{reg}$ using all previous masks {$R_1, R_2,...,R_{j-1}$} according to Eq. 2.
> &nbsp;&nbsp;&nbsp;&nbsp;&nbsp;&nbsp;&nbsp;&nbsp;&nbsp;&nbsp;&nbsp;&nbsp;Set total loss $L_{total} = L_{lm} + L_{reg}$
> &nbsp;&nbsp;&nbsp;&nbsp;&nbsp;&nbsp;&nbsp;&nbsp;**else**
> &nbsp;&nbsp;&nbsp;&nbsp;&nbsp;&nbsp;&nbsp;&nbsp;&nbsp;&nbsp;&nbsp;&nbsp;Set total loss $L_{total} = L_{lm}$
> &nbsp;&nbsp;&nbsp;&nbsp;&nbsp;&nbsp;&nbsp;&nbsp;Update model parameters to minimize $L_{total}$
> &nbsp;&nbsp;&nbsp;&nbsp;Compute the regularization mask $R_j$ for the current task
> &nbsp;&nbsp;&nbsp;&nbsp;Merge $LoRA_j$ into $M_{j-1}$ to form $M_j$
>
> ---
>
> Here, the language modeling loss $L_{lm}$ refers to the next-token prediction loss used by the base MLLM.
> After learning each task $j$, we evaluate performance on all learned tasks using the updated model $M_j$.

---

### Official Review · Reviewer_iTsR · 2025-03-08

**Overall Recommendation:** 4

**Summary:**

This manuscript delves into the area of Multimodal Continual Instruction Tuning (MCIT). It contributes by differentiating between two forms of catastrophic forgetting: *superficial forgetting* and *essential forgetting*. Superficial forgetting is defined as the forgetting of the response style, while essential forgetting denotes the loss of knowledge. To tackle these problems, the manuscript introduces two corresponding solutions.  Firstly, it introduces the Answer Style Diversification (ASD) paradigm. This paradigm unifies questions within each task into five predefined styles. By doing so, it alleviates the variability in question domains, the main cause of superficial forgetting. Secondly, the manuscript presents RegLoRA. This method restricts modifications to the elements in LoRA’s weight update matrices that are highly relevant to previously learned knowledge to prevent essential forgetting.
The experimental results show that the proposed overall approach, SEFE, attains state-of-the-art performance. The effectiveness of both the ASD and RegLoRA techniques is also validated.

**Claims And Evidence:**

This manuscript's main claims are well-supported by experimental evidence, rendering them convincing. However, I take issue with a statement in the Introduction (line 30-31), which reads: "Training these models typically involves two main phases: pre-training and instruction tuning". Based on my knowledge, the training of Multimodal Large Language Models (MLLMs)/Large Language Models (LLMs) usually encompasses more than two stages. It may also incorporate a Preference Optimization/Reinforcement Learning phase. Although this statement does not substantially undermine the manuscript's contributions, I still suggest that the authors revise it to ensure accuracy.

**Essential References Not Discussed:**

Since the RegLoRA proposed in this manuscript is a variant of LoRA, I suggest adding an introduction to other LoRA variants in Appendix B (Related Works) for differentiation. For example, the following papers:

[1]	Zhang, Qingru, et al. "AdaLoRA: Adaptive budget allocation for parameter-efficient fine-tuning." arXiv preprint arXiv:2303.10512 (2023).

[2]	Hayou, Soufiane, Nikhil Ghosh, and Bin Yu. "Lora+: Efficient low rank adaptation of large models." arXiv preprint arXiv:2402.12354 (2024).

[3]	Liu, Shih-Yang, et al. "Dora: Weight-decomposed low-rank adaptation." arXiv preprint arXiv:2402.09353 (2024).

**Experimental Designs Or Analyses:**

-	**Comparison with existing methods**: The authors utilize the public CoIN benchmark and the CoIN-ASD benchmark proposed in this work. The evaluation includes two components: Truth Alignment (TA) and Knowledge Capability (KC). Each component encompasses the accuracy of all tasks, along with four aggregate metrics: Mean Fine-tune Accuracy (MFT), Mean Final Accuracy (MFN), Mean Average Accuracy (MAA), and Backward Transfer (BWT). These metrics are quite comprehensive.
-	**Ablation studies**: The ablation experiments cover the validation of ASD, RegLoRA, and the selection of hyperparameters. The metrics employed are the aggregate metrics of TA and KC. The authors present an analysis for each experiment, which is reasonable overall.

**Methods And Evaluation Criteria:**

The proposed SEFE method holds value in the continuous improvement of MLLMs to enable their adaptation to new demands. Additionally, the introduced CoIN-ASD benchmark can be useful for evaluating essential forgetting within the MCIT domain.

**Other Comments Or Suggestions:**

The Related Works section is currently located in the appendix. This makes it hard to quickly grasp the distinctiveness of this manuscript without referring to the appendix. I suggest adding a concise Related Works section within the main text. The introduction to LoRA in the RegLoRA section could be moved there. A more in-depth review of related works can still be retained in the appendix.

**Other Strengths And Weaknesses:**

The primary strength of this work lies in its definition of superficial forgetting and essential forgetting. The experimental results and analysis support the need to investigate these phenomena separately. Additionally, the proposed method is logically sound and demonstrates strong performance.
However, several weaknesses remain, in addition to the previously mentioned issues (lack of rigor in one statement and the absence of a review of other LoRA variants):

-	**Limited dataset diversity**: Although this study includes both CoIN and CoIN-ASD benchmarks, these datasets are essentially derived from the same source, which may weaken the generalizability of the experimental results. That said, given that CoIN is the only publicly available MCIT benchmark, this limitation is somewhat understandable.
-	**Incomplete definition of the learning objective**: In line 272-274, the authors state that “L_reg is added to the original loss of the base MLLM to form the complete learning objective.” However, the manuscript does not clearly define the original loss or the overall learning objective. This omission reduces the study’s reproducibility.

**Questions For Authors:**

I am curious about one aspect: In the superficial forgetting scenario where the model responds to any question with a bounding box (e.g., Fig. 1(d)), consider a situation where both the question and the answer contain objects, yet these objects differ. Will the bounding box indicate the object in the question or the one in the answer? For instance, if the question is “What is to the left of the sheep?” and the correct answer is “A cow”, when superficial forgetting occurs, does the returned bounding box point to the sheep or the cow? Or perhaps it points to neither? I think this aspect is valuable for exploring the essential forgetting.

**Relation To Broader Scientific Literature:**

The RegLoRA proposed in this manuscript is a regularization-based continual learning scheme. It has certain relevance to prior methods like EWC [1] and PODNet [2]. These strategies reduce the magnitude of model updates following specific rules to alleviate forgetting.

[1]	Kirkpatrick, James, et al. "Overcoming catastrophic forgetting in neural networks." Proceedings of the national academy of sciences (2017).

[2]	Douillard, Arthur, et al. "Podnet: Pooled outputs distillation for small-tasks incremental learning." ECCV, 2020.

**Theoretical Claims:**

This manuscript primarily focuses on method design and evaluates its effectiveness through experimental observation and analysis. It does not include significant theoretical discussion or mathematical derivation. Therefore, this section is not applicable.

---

> ### Author Rebuttal · Authors · 2025-03-29
>
> Thank you for your thorough review and for recognizing the value of our contributions. Below, we address your comments in detail:
>
> ### **Training Stages of MLLMs**
> Thank you for pointing this out. Indeed, the training process of MLLMs/LLMs often involves more than two stages. We will revise the sentence to: "Training these models typically involves multiple phases, with pretraining and instruction tuning being two crucial ones."
>
> ### **Introduction of LoRA Variants**
> Thank you for your constructive feedback. We will include an overview of LoRA variants, such as those you mentioned: AdaLoRA, which adaptively allocates the parameter budget across weight matrices based on their importance scores; LoRA+, which assigns different learning rates to matrices A and B to enhance performance in settings with large embedding dimensions; and DoRA, which decomposes pretrained weights into magnitude and direction components, applying LoRA solely to the direction component to reduce the number of trainable parameters. While these methods are not primarily designed to mitigate catastrophic forgetting and may not be directly applicable to MCIT, some of their underlying principles may offer valuable insights. We have therefore included a discussion of these approaches in a separate subsection within the Related Work section for readers’ reference.
>
> ### **Incomplete Definition of the Learning Objective**
> The original loss function is the standard language modeling loss, *i.e.*, next-token prediction loss, consistent with the base model (LLaVA-1.5). It is defined as:
> $$L_{lm} = - \sum_{i=1}^{L} \log P(x_i | X_v, X_\text{instruct,<i}, X_{a,<i}),$$
> where $L$ is the length of the answer, $X_v$ denotes the visual input tokens, $X_\text{instruct,<i}$ and $X_{a,<i}$ represent the instruction and answer tokens prior to the current prediction token $x_i$, respectively.
> During training on the first task, the objective is solely this language modeling loss. For subsequent tasks, the total loss becomes a combination of the language modeling loss and the regularization loss $L_{reg}$ as defined in Eq. 2 of the paper:
> $$L_{total} = L_{lm} + L_{reg}.$$
> We will incorporate this complete formulation and explanation into the revised version.
>
> ### **Adding a Related Works Section in the Main Text**
> Thank you for the helpful suggestion. As recommended, we will add a concise Related Works section to the main text, include the introduction to LoRA and a brief review of MCIT works within it, and retain the more detailed review of related literature in the appendix.
>
> ### **Bounding Box Responses under *Superficial Forgetting***
> Your idea is compelling, but it is challenging to identify samples in our existing dataset that match the scenario you described—where both the question and the answer contain an object present in the image. To investigate this, we manually constructed a few examples. Our observations indicate that the model’s predicted bounding boxes tend to align with the object mentioned in the question, rather than the one in the answer. This behavior reflects an inductive bias introduced by the grounding task, in which training samples typically require the model to localize the object referenced in the question. These findings suggest that *superficial forgetting* may lead the model to over-rely on task-specific biases, hindering its ability to adapt to the actual demands of the current query. Rather than interpreting the intent of the question, the model appears to default to habitual responses based on prior training. If the model had even partially understood the question, we would expect it to at least identify the object mentioned in the answer. This further illustrates how *superficial forgetting* can obscure the model’s ability to adjust its behavior based on contextual demands. Thank you for your insightful question.

---

### Official Review · Reviewer_NhXB · 2025-03-10

**Overall Recommendation:** 3

**Summary:**

This paper introduces SEFE for multimodal continual instruction tuning. Within SEFE, an ASD paradigm is proposed to eliminate superficial forgetting by converting all questions into five unified question types. The authors also create a CoIN-ASD benchmark by applying ASD to the public CoIN benchmark, which can be used to assess essential forgetting. Moreover, a Reglora is proposed to eliminate essential forgetting by adding regularization to key elements of lora weight.

**Claims And Evidence:**

In general, all major claims are proven by experiments. I didn’t find any major claims that is obviously not convincing.

**Essential References Not Discussed:**

None. This paper reviewed sufficient literatures.

**Experimental Designs Or Analyses:**

Yes. The experimental designs are basically sound and fair. They involve a reasonable setting and maintain same for all compared and ablation methods (incrementally training CoIN tasks one-by-one).

**Methods And Evaluation Criteria:**

Yes. SEFE makes sense for continually adapting MLLMs to new tasks, which can be helpful in dynamic scenarios. The evaluation criteria are also solid, covering many aspects.

**Other Comments Or Suggestions:**

Figure 3 is not very clear. It is recommended to use a plus sign to indicate the combination of R1 of R2.

**Other Strengths And Weaknesses:**

# Strengths:
1. The performance is satisfactory.
2. Sufficient experimental results, including comparisons and ablation studies, are reported.

# Weaknesses:
I have some concerns regarding the ASD paradigm:
1. When you generate Yes/No questions and MCQs, you use InternVL2 to generate distractors. However, for some cases the answer domain of a task is limited (for example, in ImageNet, the answer domain contains only 1000 category names), this generation step is not necessary, because distractors can be simply selected from the answer domain (excluding the correct answer). This issue is not sufficiently discussed and defined in the paper.
2. InternVL2 is used to generated explanations in ASD. However, my concern is that, if InternVL2 can generate accurate explanations, doesn’t that mean InternVL2 can totally understand and solve the task? Why we need to continually train another MLLM?

**Questions For Authors:**

Please refer to the weaknesses and comments in previous two sections.

**Relation To Broader Scientific Literature:**

The definition of superficial and essential forgetting in this paper seems to be expandable to continual learning of LLM and other large-scale models, and the proposed solutions may also be applicable to these and related areas.

**Theoretical Claims:**

This paper doesn’t have theoretical claims.

---

> ### Author Rebuttal · Authors · 2025-03-29
>
> Thank you for reviewing our paper and for your positive feedback on the performance and experimental validation of our proposed method. Below, we address your concerns in detail:
>
> ### **Unnecessary Use of InternVL2 in Certain Distractor Generation Scenarios**
> As you correctly noted, using InternVL2 for distractor generation is unnecessary in some cases. In the CoIN dataset, this applies to the ImageNet and Grounding tasks. Accordingly, we did not employ InternVL2 for these tasks. For ImageNet, distractors are randomly sampled from the class list excluding the ground-truth one. For Grounding, we generate random bounding boxes with Intersection-over-Union (IoU) less than 0.5 with respect to the ground-truth box.
> While this special handling is briefly mentioned in Appendix C.2, a formal definition was not provided. Following your suggestion, we will incorporate a more rigorous definition based on the characteristics of the answer domain, enabling this issue to be addressed not only in CoIN but also in other benchmarks and real-world applications.
>
> ### **Capability of InternVL2 to Understand and Solve the Task**
> The ability of InternVL2 to generate explanations does not imply a full understanding of the task. This is because the model is provided with the ground-truth answer during explanation generation, eliminating the need for answer reasoning. Consequently, the model’s task is only generating an explanation conditioned on the correct answer, which is significantly easier.
> To validate this point, we conducted experiments evaluating InternVL2-26B on the eight tasks in CoIN. InternVL2-26B achieves an average accuracy of 53.76%, whereas our SEFE model attains 58.57% after continual learning across all tasks. Notably, SEFE is based on a 7B model, considerably smaller than InternVL2-26B. These results demonstrate that continual training of an MLLM is meaningful because it can yield superior performance with fewer inference resources.
>
> ### **Modification of Fig. 3**
> Thank you for the suggestion. We will revise Fig. 3 to include a plus sign, clearly indicating the combination method of $R_1$ and $R_2$.

---

> > ### Comment · Reviewer_NhXB · 2025-04-07
> >
> > Thank you for the authors’ responses, which have addressed my previous concerns. I have no further comments. Having seen that other reviewers are also supportive, I will maintain my support as well.

---

> > > ### Author Response · Authors · 2025-04-09
> > >
> > > Thank you very much for your support and the positive feedback on our paper.

---

### Official Review · Reviewer_Hv34 · 2025-03-14

**Overall Recommendation:** 3

**Summary:**

This paper introduces SEFE (Superficial and Essential Forgetting Eliminator), a novel framework for Multimodal Continual Instruction Tuning (MCIT), which aims to prevent catastrophic forgetting in multimodal models. The authors identify two distinct types of forgetting in MCIT”: superficial forgetting and essential forgetting. To mitigate these issues, the paper proposes two key techniques: Answer Style Diversification (ASD) and RegLoRA for these two types of forgetting respectively. The proposed SEFE method achieves state-of-the-art performance in MCIT benchmarks, including CoIN and the CoIN-ASD introduced by the paper, demonstrating its effectiveness in reducing both superficial and essential forgetting.

## Update After Rebuttal
My original assessment was supportive, so I will maintain my current score.

**Claims And Evidence:**

Yes

**Essential References Not Discussed:**

The paper lacks a discussion of and comparison with related works on understanding catastrophic forgetting such as [SP’2024].

[SP’2024] Zheng, J., Cai, X., Qiu, S., & Ma, Q. (2025). Spurious Forgetting in Continual Learning of Language Models. arXiv preprint arXiv:2501.13453.

**Experimental Designs Or Analyses:**

Yes

**Methods And Evaluation Criteria:**

Yes

**Other Comments Or Suggestions:**

No additional comments.

**Other Strengths And Weaknesses:**

Strengths:
1. This paper proposes and provides evidence to support a new perspective for understanding catastrophic forgetting.

Weaknesses:
1. Lack of comparison and discussion of a line of works for understanding catastrophic forgetting such as [SP’2024].

**Questions For Authors:**

1. Can the authors clarify the difference between their work and a similar work, [SP’2024]?

**Relation To Broader Scientific Literature:**

Catastrophic forgetting is a key challenge in continual learning. This paper offers a new perspective by categorizing it into two types: superficial forgetting and essential forgetting.

**Theoretical Claims:**

N/A

---

> ### Author Rebuttal · Authors · 2025-03-29
>
> We appreciate your time and effort in reviewing our paper, as well as your recognition of our contributions. Below, we respond to your comments in detail:
>
> ### **Comparison with the [SP’2024] Paper**
>
> #### **Differences between the Two Papers**
>
> Thank you for pointing out this related work. This study indeed shares our goal of understanding catastrophic forgetting, and its concept of *spurious forgetting* is in some respects similar to our definition of *superficial forgetting*. However, there are several key differences between the two studies.
>
> - *Superficial forgetting* refers to cases where the model fails to generate responses in the expected format, indicating a loss of response style. In such situations, it remains unclear whether the underlying knowledge has actually been forgotten. In contrast, *spurious forgetting* describes scenarios in which the model loses task alignment without any genuine loss of knowledge. Since *spurious forgetting* can be easily recovered, it is referred to as "spurious", whereas *superficial forgetting* does not emphasize recoverability. Thus, these two concepts are distinct.
>
> - The [SP’2024] paper focuses primarily on the analysis of forgetting, examining changes in gradients and feature principal components during forgetting. Although it proposes a mitigation strategy—*Freeze*, which freezes a few bottom layers—this approach is simple, is not the core contribution of the paper, and shows limited effectiveness (*e.g.*, ~60% forgetting rate). In contrast, our work focuses on practical solutions. We propose Answer Style Diversification (ASD) and RegLoRA to address both *superficial forgetting* and *essential forgetting* via data reconstruction and weight update constraints. On the CoIN benchmark, our method reduces the average forgetting rate to 10.45%, substantially lower than the 60% reported in the [SP’2024] paper. While differences in datasets limit direct comparability, the trend highlights the greater effectiveness and practicality of our approach. By contrast, *Freeze* serves more as an analytical tool of *superficial forgetting* than a standalone solution.
>
> - In addition to *superficial forgetting*, our study also examines *essential forgetting*, where the underlying knowledge is genuinely lost. In contrast, the [SP’2024] paper focuses exclusively on *spurious forgetting* that involves no knowledge degradation. This highlights another fundamental difference between the two studies.
>
> - Finally, the [SP’2024] paper focuses on LLMs, whereas our work addresses forgetting in MLLMs, marking a clear difference in research scope.
>
> #### **Performance Comparison**
> We also evaluated *Freeze* on our benchmark, with the following results:
> | Method  | MFT   | MFN   | MAA   | BWT    |
> |---------|-------|-------|-------|--------|
> | FFT     | 65.87 | 35.45 | 36.73 | -30.42 |
> | Freeze  | 66.04 | 39.53 | 37.60 | -26.51 |
> | Ours    | **69.02** | **58.57** | **63.04** | **-10.45** |
>
> Here, FFT refers to full-parameter fine-tuning and serves as the baseline. While *Freeze* outperforms the baseline, it consistently underperforms compared to our method across all metrics. This is likely because *Freeze* is not well-suited for Multimodal Continual Instruction Tuning (MCIT), as it was not originally designed for this setting. Additionally, as discussed earlier, *Freeze* functions more as a supplementary analytical tool, so its performance may not be very satisfactory as a dedicated continual learning method.
>
> ### **Comparison with Other Papers**
>
> In addition to the [SP’2024] paper, we identified several other studies that investigate catastrophic forgetting. For example, [1] shows that catastrophic forgetting during LLM fine-tuning becomes more pronounced as the loss landscape sharpens, suggesting a strong positive correlation between sharpness and forgetting. [2] argues that in MLLMs, catastrophic forgetting arises as fine-tuning shifts the model’s focus from general visual-text alignment to dataset-specific overfitting, resulting in performance degradation even when the vision encoder is frozen. In contrast to these approaches, our work proposes decomposing catastrophic forgetting in MCIT into two components—*superficial forgetting* and *essential forgetting*—and addresses them separately, offering a new perspective on the problem.
>
> As suggested, we will include a comparison between our work and other related works (such as [SP’2024], [1], [2]) for understanding catastrophic forgetting in the revised version.
>
> [1] [EMNLP 2024] Revisiting Catastrophic Forgetting in Large Language Model Tuning
> [2] [CPAL 2024] Investigating the Catastrophic Forgetting in Multimodal Large Language Models

---

> > ### Comment · Reviewer_Hv34 · 2025-04-09
> >
> > Thank you to the author for addressing my concerns. I will maintain my assessment as it has been supportive.

---

> > > ### Author Response · Authors · 2025-04-09
> > >
> > > Thank you so much. Your support is truly appreciated.

---

### Decision · Program_Chairs · 2025-05-01

**Decision:**

Accept (poster)

**Comment:**

This paper presents SEFE, a framework for Multimodal Continual Instruction Tuning (MCIT), which introduces a two-fold categorization of forgetting: superficial and essential. To address these, the authors propose the Answer Style Diversification (ASD) paradigm to prevent format-related degradation (superficial forgetting), and RegLoRA, a regularization-based method to mitigate knowledge degradation (essential forgetting). The paper is complemented by the introduction of the CoIN-ASD benchmark for better evaluation of MCIT challenges.

The reviewers appreciated the paper's conceptual clarity, practical impact, and strong empirical results, with multiple reviewers highlighting the distinction between the two types of forgetting as both novel and useful. The proposed methods were shown to outperform baselines across multiple metrics and ablation settings. While a few reviewers raised minor concerns—such as limited dataset diversity and comparisons with other LoRA variants—these were not deemed critical, and the overall consensus was positive.